# Proteasome inhibition triggers tissue-specific immune responses against different pathogens in *C. elegans*

**Manish Grover[1], Spencer S. Gang[2¤], Emily R. Troemel[2], Michalis Barkoulas[1]***

**1** Department of Life Sciences, Imperial College, London, United Kingdom, **2** School of Biological Sciences, University of California, San Diego, La Jolla, California, United States of America

¤ Current address: Department of Molecular Biology, Colorado College, Colorado Springs, Colorado, United States of America

* m.barkoulas@imperial.ac.uk

**Data Availability Statement:** All relevant data are within the paper and its Supporting Information files. All RNAseq files are available from the NCBI GEO database (accession number GSE241087).

## Abstract

Protein quality control pathways play important roles in resistance against pathogen infection. For example, the conserved transcription factor SKN-1/NRF up-regulates proteostasis capacity after blockade of the proteasome and also promotes resistance against bacterial infection in the nematode *Caenorhabditis elegans*. SKN-1/NRF has 3 isoforms, and the SKN-1A/NRF1 isoform, in particular, regulates proteasomal gene expression upon proteasome dysfunction as part of a conserved bounce-back response. We report here that, in contrast to the previously reported role of SKN-1 in promoting resistance against bacterial infection, loss-of-function mutants in *skn-1a* and its activating enzymes *ddi-1* and *png-1* show constitutive expression of immune response programs against natural eukaryotic pathogens of *C. elegans*. These programs are the oomycete recognition response (ORR), which promotes resistance against oomycetes that infect through the epidermis, and the intracellular pathogen response (IPR), which promotes resistance against intestine-infecting microsporidia. Consequently, *skn-1a* mutants show increased resistance to both oomycete and microsporidia infections. We also report that almost all ORR/IPR genes induced in common between these programs are regulated by the proteasome and interestingly, specific ORR/IPR genes can be induced in distinct tissues depending on the exact trigger. Furthermore, we show that increasing proteasome function significantly reduces oomycete-mediated induction of multiple ORR markers. Altogether, our findings demonstrate that proteasome regulation keeps innate immune responses in check in a tissue-specific manner against natural eukaryotic pathogens of the *C. elegans* epidermis and intestine.

## Introduction

The evolutionary history of *Caenorhabditis elegans* is shaped by various biotic interactions in its natural habitat [1]. These interactions include beneficial and pathogenic microbes, which trigger a diverse set of responses in *C. elegans* [1]. Studying *C. elegans* under some conditions

**Funding:** This work was supported by the Wellcome Trust, UK (219448/Z/19/Z to MB) and the Biotechnology and Biological Sciences Research Council, UK (BB/X001865/1 to MB and ERT). The funders had no role in study design, data collection and analysis, decision to publish, or preparation of the manuscript.

**Competing interests:** The authors have declared that no competing interests exist.

**Abbreviations:** BTZ, bortezomib; EMS, ethyl methanesulfonate; IPR, intracellular pathogen response; ORR, oomycete recognition response; PRAAS, proteasome-associated autoinflammatory syndrome; smFISH, single molecule fluorescence in situ hybridization; UPR, unfolded protein response.

encountered in its natural environment can provide novel insights, such as new functions for genes lacking functional annotation in the genome [1]. For example, the identification of oomycetes as natural pathogens of *C. elegans* revealed a previously uncharacterized family of *chitinase-like (chil)* genes as immune response effectors, which can modify cuticle properties to prevent oomycete attachment and consequently infection [2]. Most *chil* genes are not expressed in lab culture conditions and can only be induced upon pathogen recognition together with a broader subset of genes previously described as the oomycete recognition response (ORR) [3]. *C. elegans* can similarly mount specific responses against other natural pathogens as well, for example, the intracellular pathogen response (IPR) against microsporidia and the Orsay virus [4,5].

While the nematode is likely to use its sensory capabilities to detect pathogens and activate pathogen-specific responses [6], it also uses surveillance immunity as a broad way to tackle pathogenic attacks [7]. Pathogens can hijack the host cellular machinery to support their growth and development, and in doing so, they may disrupt core cellular processes, such as transcription, translation, protein turnover, and mitochondrial respiration. As a result, disruption to these processes can be sensed as a pathogenic attack leading to activation of immune responses [8–10]. For example, inhibition of translation by *Pseudomonas aeruginosa* exotoxin A leads to activation of immune response genes through transcription factors such as ZIP-2 and CEBP-2 [8,9,11]. Disruption of mitochondrial proteostasis by *P. aeruginosa* also induces immune responses [12,13]. Similarly, RNAi and microbial toxin perturbation of various core cellular processes induces detoxification enzymes and aversion behavior in *C. elegans* [10]. Likewise, the IPR can be induced either by inhibition of the purine salvage pathway as seen in *pnp-1* mutants [14,15] or by blockade of the major protein degradation machinery in the cell, the proteasome [4,16].

A major player regulating proteostasis upon proteasome blockade is the conserved transcription factor SKN-1/NRF [17,18]. One of its isoforms, SKN-1A (NRF1/NFE2L1 in humans), is normally localized in the endoplasmic reticulum membrane where it is glycosylated and then targeted for proteasomal degradation (Fig 1A) [19]. However, upon proteasome blockade, the glycosylated isoform escapes degradation and is edited by PNG-1 (NGLY1 in humans), a conserved N-glycanase that converts the N-glycosylated asparagine residues to aspartic acid, and the edited protein then translocates to the nucleus [19]. Inside the nucleus, DDI-1 (DDI2 in humans), a conserved aspartic protease, cleaves the N-terminal region of the protein, and this processed SKN-1A is now activated to up-regulate expression of proteasomal subunits to increase proteostasis capacity in a bounce-back response (Fig 1A) [19–21]. In addition to its role in promoting proteostasis capacity, SKN-1 is also required to respond to oxidative stress [22] and promotes resistance to bacterial pathogens, such as the human pathogens *P. aeruginosa* and *Enterococcus faecalis* [23–25].

Here, we show that, in contrast to previous studies demonstrating a protective role for SKN-1 in promoting resistance against bacterial pathogens, mutants in the SKN-1A-driven proteasome surveillance pathway result in constitutive expression of ORR and IPR and consequently exhibit resistance against infection by oomycetes and microsporidia. Rescue of *skn-1a* specifically in the epidermis or the intestine was sufficient to restore wild-type levels of infection by oomycetes and microsporidia, respectively. Moreover, we report that the ORR/IPR genes induced in common in these programs, are also regulated through the proteasome and can be induced in distinct tissues depending on the exact trigger. Notably, we show that by increasing proteasome function, there is a partial inhibition of oomycete-mediated induction of ORR markers, suggesting that the response to pathogens may directly involve stabilization of factors normally degraded by the proteasome. Therefore, our findings, together with other studies in flies and mammals [26–29], highlight how proteasome regulation keeps a range of innate immune responses in check in a tissue-specific manner.

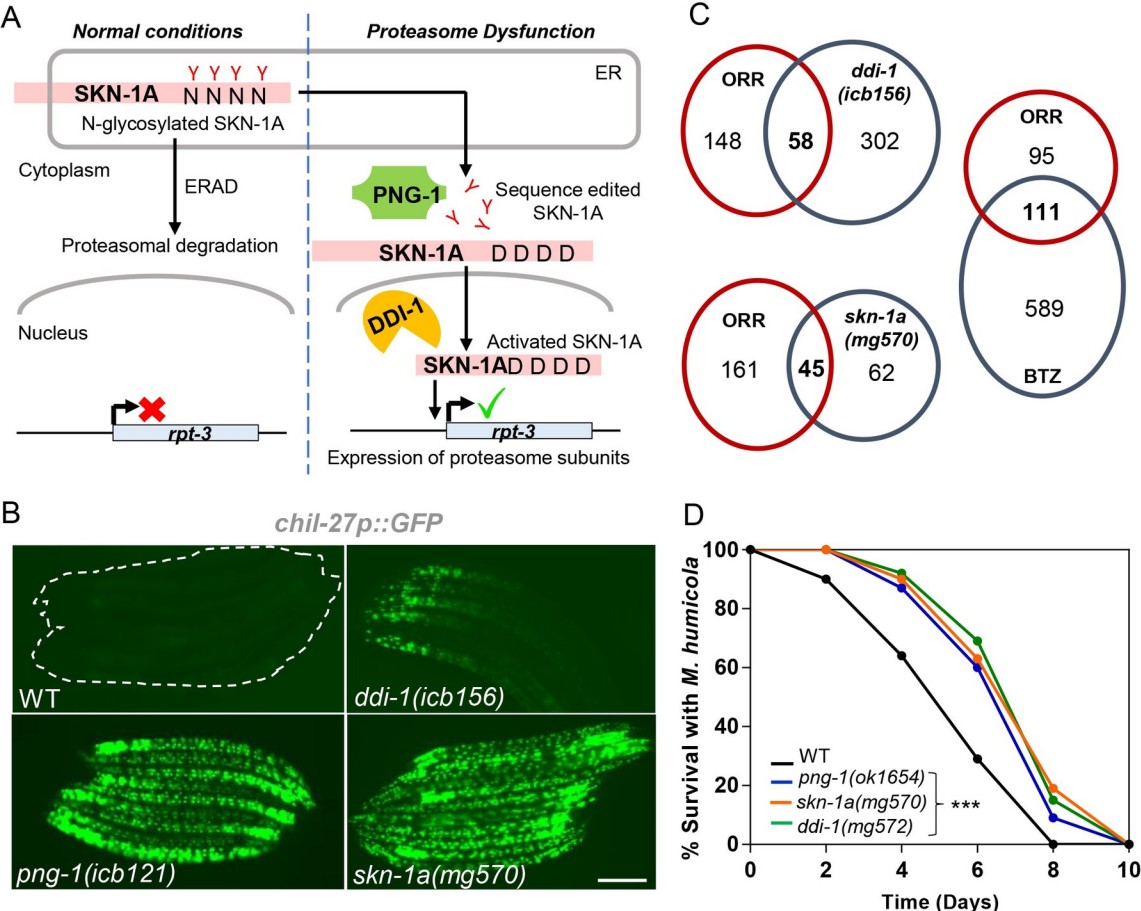

**Fig 1. Proteasome impairment activates ORR in *C. elegans*.** **(A)** Schematic showing the proteasome surveillance pathway and the stepwise activation of SKN-1A for transcription of proteasomal subunits. **(B)** L4 stage *C. elegans* showing constitutive *chil-27p::GFP* expression in *ddi-1(icb156)*, *png-1(icb121)*, and *skn-1a(mg570)* mutant animals. Note that *skn-1a* and *png-1* show a stronger, full body activation of the *chil-27p::GFP* marker in comparison to *ddi-1* mutants, which might reflect the fact that *png-1* and *skn-1a* are essential components of the proteasome surveillance pathway, while *ddi-1* has been shown to be dispensable [19]. Scale bar is 100 μm. **(C)** Venn comparisons showing significant overlap between up-regulated genes in the transcriptome of *ddi-1(icb156)* and *skn-1a(mg570)* mutant, and BTZ-treated animals with ORR (RF 13.8, $p < 6.171e-50$ for comparison with *ddi-1*; RF 13.6, $p < 5.518e-101$ for comparison with BTZ and RF 36.0, $p < 1.249e-59$ value for comparison with *skn-1a*). **(D)** *ddi-1(mg572)*, *png-1(ok1654)*, and *skn-1a(mg570)* mutant animals exhibit reduced susceptibility to infection by *M. humicola* as compared to WT *C. elegans* ($n = 60$ per condition, performed in triplicates, $p < 0.001$ based on log-rank test, a representative graph for one of the 3 replicates is shown). The numerical data for all 3 replicates is available in Supporting information S1 Data. BTZ, bortezomib; ORR, oomycete recognition response.

## Results

### Mutants in the proteasome surveillance pathway show constitutive activation of the ORR

We performed a chemical mutagenesis screen on animals carrying the *chil-27p::GFP* reporter, which is not expressed in wild-type animals under standard growth conditions, but is strongly induced upon recognition of oomycete pathogens [2,30]. We obtained several independent mutants with constitutive epidermal expression of *chil-27p::GFP*, either in a graded manner with maximum GFP expression in the head region as previously reported [2], or strong expression throughout the body in late larval stages and early adults (Fig 1B). Three mutations were mapped to *ddi-1* and *png-1*, 2 genes known to play a role in the proteasome surveillance

pathway by regulating SKN-1A [18,19] (Figs 1A and S1A). To determine if the non-synonymous *ddi-1(icb156)* mutation isolated from the screen represented a gain or loss-of-function allele, we tested existing strong loss-of-function *ddi-1* mutants carrying either a deletion within the gene (*mg571*) or a mutation in the active site of the protein (*mg572*) [18]. In both cases, we found similar *chil-27p*::GFP induction (S1B Fig), confirming that *ddi-1(icb156)* represents a loss-of-function allele. Similarly, the recovery of a frameshift and a nonsense mutation in *png-1* suggested that constitutive *chil-27p*::GFP expression is likely attributed to loss-of-function of the gene, which was further confirmed using the independently derived *png-1(ok1654)* deletion allele [31] (S1B Fig).

As PNG-1 and DDI-1 are required to activate SKN-1A, we analyzed *chil-27*::GFP expression upon loss-of-function of *skn-1a*. Here, we found constitutive expression throughout the body of the animal (Fig 1B). The response was also recapitulated by *skn-1* RNAi (S1B Fig), which is known to affect both *skn-1a* and *skn-1c* isoforms as the entire *skn-1c* sequence is shared by *skn-1a* (S2A Fig). Because PNG-1 is required specifically for activation of the SKN-1A isoform and SKN-1C does not undergo sequence editing [19], the involvement of SKN-1C in activating *chil-27p*::GFP seemed unlikely. However, to test this possibility, we performed *wdr-23* RNAi on *skn-1a(mg570)* mutant animals. WDR-23 is a WD40 protein known to specifically suppress SKN-1C function, inhibition which is released, for example, under oxidative stress to allow SKN-1C activation [32,33]. We found that *wdr-23* RNAi did not suppress *chil-27p*::GFP induction in *skn-1a(mg570)* mutants, while it activated a *gst-4p*::tdTomato reporter in wild type as expected (S2B Fig). Furthermore, we treated *skn-1a(mg570)* animals with *skn-1* RNAi to address the possibility of *skn-1c* isoform contributing to *chil-27p*::GFP induction in the absence of *skn-1a*. However, *skn-1* RNAi did not have any effect on the expression of *chil-27p*::GFP in *skn-1a* mutants (S2B Fig). Altogether, these results suggest that the *skn-1c* isoform does not regulate expression of *chil-27p*::GFP, which is induced specifically by the loss of the *skn-1a* isoform.

Mutants in *skn-1a* have reduced proteasome subunit gene expression and, as a result, experience proteasome dysfunction [34]. Reduced proteasomal subunit gene expression may affect the induction of immune programs through non-proteolytic roles that have recently been ascribed to proteasomal subunits. If this were the case, then induction would not be affected by proteasome impairment via bortezomib (BTZ) drug treatment [35]. To determine if impairment of proteasome function can induce the ORR, we performed RNAseq analysis of *ddi-1(icb156)* mutants versus wild-type controls and compared the set of up-regulated genes with those previously reported for inhibition of proteasomal activity by BTZ drug treatment [36], oomycete recognition response [3], and *skn-1a* loss-of-function [36]. A significant overlap was identified between all these datasets (Figs 1C and S1C, and S1 Table). Wild-type animals treated with high dose of BTZ show induction of *chil-27* as previously described [16], and this induction is rapid at the *chil-27* mRNA level within 15 min post exposure to the drug as opposed to 1 h required upon exposure to oomycete extract (S3A and S3B Fig), which further suggested a link between the induction of ORR and proteasome dysfunction. The physiological consequence of activated ORR was revealed by an infection assay where *ddi-1(mg572)*, *png-1 (ok1654)*, and *skn-1a(mg570)* mutants showed enhanced survival in the presence of the oomycete pathogen *Myzocytiopsis humicola* (Fig 1D). The enhanced survival phenotype was recapitulated both by *skn-1* RNAi treated WT or *skn-1a(mg570)* animals (S2C Fig), as well as BTZ-treated animals (S3C Fig). These findings are consistent with ORR induction in *skn-1a* mutants being protective against oomycete infection and associated with loss of proteasomal activity, as opposed to reduced proteasomal gene expression having an impact independent of proteasomal activity.

## Proteasome impairment in the epidermis is sufficient to induce the ORR

Induction of the ORR requires cross-tissue communication with chemosensory neurons likely sensing the pathogen and signaling to the epidermis where induction of *chil* genes takes place to combat the infection [3]. Having discovered that proteasome inhibition can activate the ORR, we wanted to determine whether this inhibition is required in a tissue-specific manner or not. To address this question, we performed tissue-specific rescue of *skn-1a(mg570)* mutants by expressing *skn-1a* under a *rab-3p* (pan-neuronal), *dpy-7p* (epidermal), or *vha-6p* (intestinal) promoter. We found that only the epidermal rescue of *skn-1a* repressed expression of *chil-27p::GFP* (Fig 2A). We further investigated this question by performing tissue-specific RNAi of the proteasomal subunit *rpt-5* [10,37,38], where we found that only epidermal RNAi of *rpt-5* was able to induce *chil-27p::GFP* expression (Fig 2B). We also tested the survival of tissue-specific rescued lines of *skn-1a* in the presence of *M. humicola* and found only epidermal overexpression of *skn-1a* to rescue the enhanced oomycete resistance phenotype (Fig 2C). These findings demonstrate that proteasome impairment, or rescue of proteasome surveillance function specifically in the epidermis, regulates the ORR and oomycete resistance.

Previous work has led to the identification of other epidermal regulators of ORR, namely, the receptor tyrosine kinase OLD-1 [39] that is specific to oomycete recognition, and the PALS-22/PALS-25 antagonistic paralogs [16,30], which regulate the immune response against both oomycetes and microsporidia. We thus asked whether activation of ORR upon proteasome dysfunction requires *old-1* or *pals-25*. Here, we performed *skn-1* RNAi on animals carrying a deletion in *old-1* or *pals-25*, along with wild-type animals as control, and found no difference in *chil-27p::GFP* induction (S4 Fig). These results suggest that epidermal proteasome dysfunction acts either downstream of OLD-1 and PALS-25-mediated signaling, or as a parallel trigger leading to the activation of ORR.

## Oomycete extract exposure does not cause broad proteasome dysfunction

Since epidermal proteasome dysfunction triggers the ORR, we tested whether impairment of proteasome function occurs upon exposure to oomycete extract. Proteasomal subunit expression is observed as a bounce-back response upon proteasome dysfunction [20]. Even though significant overlap was obtained between ORR, and genes up-regulated upon proteasome dysfunction (Fig 1D), none of the proteasomal components were found to be induced as a part of the ORR (Fig 3A). For example, the induction of the *rpt-3p::GFP* reporter [18] was only observed upon BTZ treatment, but not upon treatment with oomycete extract (Fig 3B). Similarly, we did not see accumulation of ubiquitylated substrates reported by a *sur-5p::UbV-GFP* marker [40] upon extract exposure, while we did observe induction of the marker upon inhibition of proteasome activity by BTZ treatment (Fig 3C). These results suggest that oomycete extract exposure is unlikely to cause broad proteasome dysfunction in *C. elegans*.

To test the potential direct involvement of proteasomal regulation for activation of the ORR upon oomycete exposure, animals constitutively expressing proteasomal subunits through a constitutively activated SKN-1A *[skn-1a(cut, 4ND)]* [19] or animals having a hyperactive proteasome (*pas-3(α3ΔN)*) [41] were treated with extract and expression of multiple ORR genes was analyzed by RT-qPCR. We found partial but significant reduction in the expression of analyzed ORR genes upon extract treatment in both cases (Fig 3D). It must be noted that none of these genes have detectable expression in the absence of extract, and their levels of expression in *skn-1a(mg570)* mutants are significantly lower compared to extract-exposed wild-type animals (S5 Fig). These results suggest that pathogen-mediated induction of the ORR epidermal signaling pathway may partly require stabilization of a factor constitutively degraded by the proteasome under uninfected conditions.

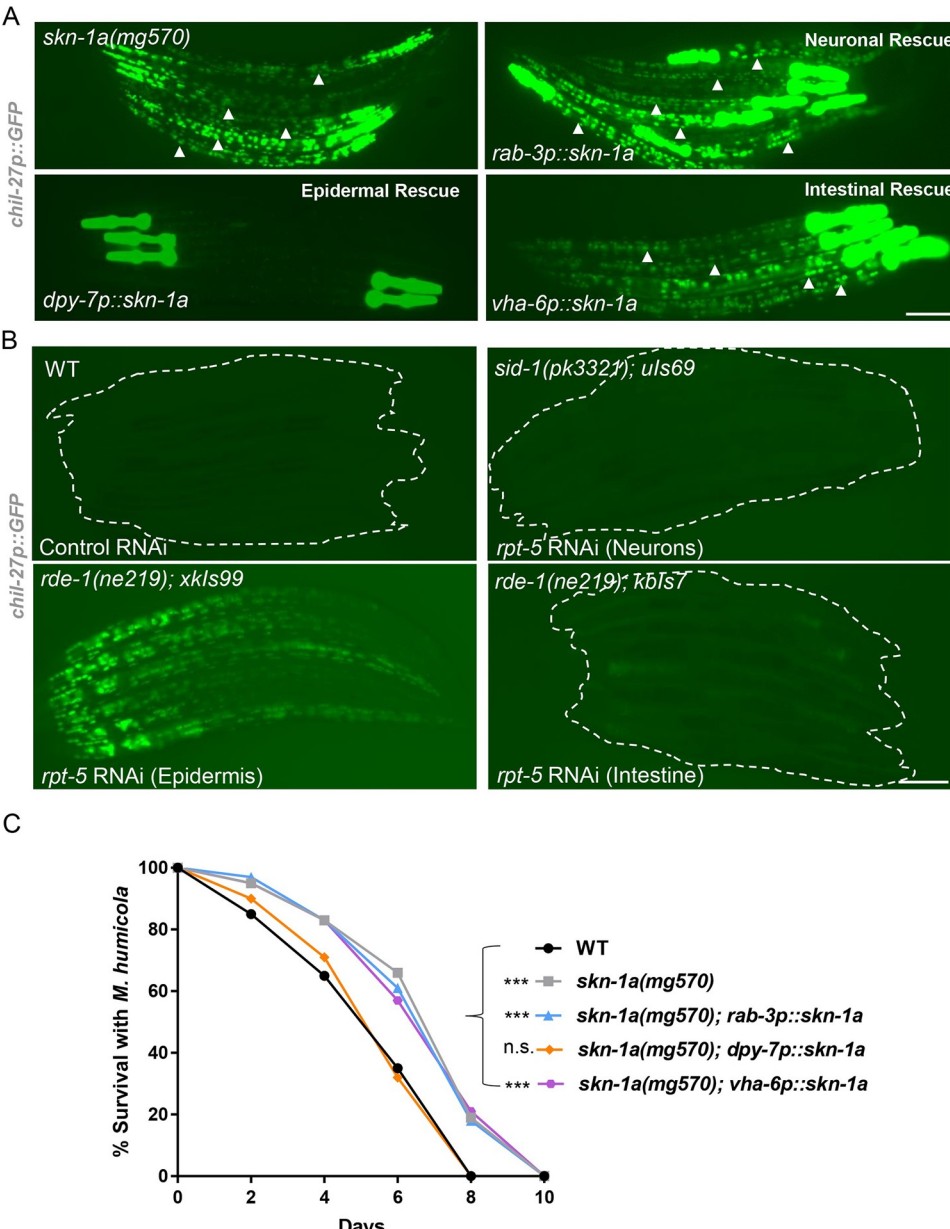

**Fig 2. Proteasome impairment in the epidermis is sufficient to activate the ORR. (A)** Epidermal (*dpy-7p*), neuronal (*rab-3p*), and intestinal (*vha-6p*) rescue of *skn-1a* function in *skn-1a(mg570)*. Note loss of GFP puncta in the body (shown by arrowheads) corresponding to *chil-27p::GFP* expression specifically upon epidermal rescue (3 independent transgenic lines analyzed, representative image shown). In all cases, *myo-2p::GFP* has been used as a co-injection marker that labels the pharynx. **(B)** Tissue-specific proteasome dysfunction induced by epidermal-specific, intestine-specific, and neuronal-enhanced RNAi of *rpt-5* (*n* > 50 per condition, performed in triplicates, representative image shown). Scale bars in A and B are 100 μm. **(C)** Survival analysis of *skn-1a(mg570)* mutants with *skn-1a* function rescued in neurons (*rab-3p::skn-1a*), epidermis (*dpy-7p::skn-1a*), and intestine (*vha-6p::skn-1a*) (*n* = 60 per condition, performed in triplicates, *p* < 0.001 based on log-rank test, a representative graph for one of the 3 replicates is shown). The numerical data for all 3 replicates is available in Supporting information S1 Data. ORR, oomycete recognition response.

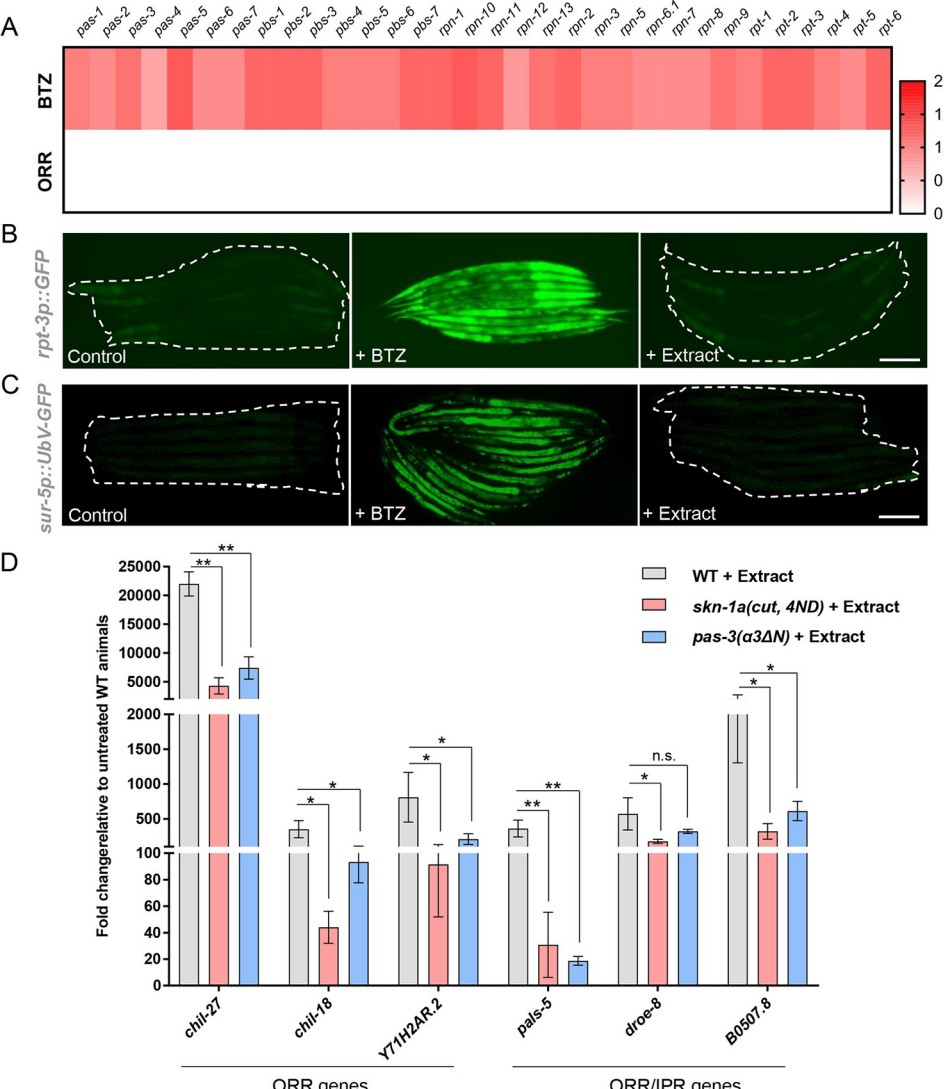

**Fig 3. Oomycete extract exposure does not cause broad proteasome dysfunction, and activation of the proteasome partially inhibits induction of ORR genes by oomycete extract. (A)** Heat map showing log2 fold change in the expression of proteasome components upon treatment with oomycete extract as opposed to BTZ treatment. None of these genes are induced as part of the ORR and are shown in white. The numerical data for the heat map is available in Supporting information S1 Data. **(B)** Induction of *rpt-3p::GFP* is observed upon BTZ treatment, but not upon extract treatment (*n* > 50, performed in triplicates, representative image shown). **(C)** Induction of *sur-5p::UbV-GFP* is observed upon BTZ treatment, but not upon extract treatment (*n* > 50, performed in triplicates, representative image shown). Scale bar in panels B and C is 100 μm. **(D)** RT-qPCR showing reduced induction of genes specific to ORR or genes in the overlap between ORR and IPR upon extract treatment in animals with constitutive expression of the activated form of SKN-1A [*skn-1a(cut, 4ND)* in *skn-1a(mg570)*] or constitutive activation of the proteasome [*pas-3 (α3ΔN)*] (**$p < 0.01$, ****$p < 0.0001$ based on unpaired *t* test in comparison to extract-treated wild type). The numerical data for all 3 replicates is available in Supporting information S1 Data. BTZ, bortezomib; IPR, intracellular pathogen response; ORR, oomycete recognition response.

## Impairment of the SKN-1A bounce-back response also leads to activation of resistance against intracellular intestinal pathogens

It is known that proteasome blockade by BTZ treatment can activate the IPR in *C. elegans* [16], which is the protective transcriptional program against intracellular intestinal pathogens like

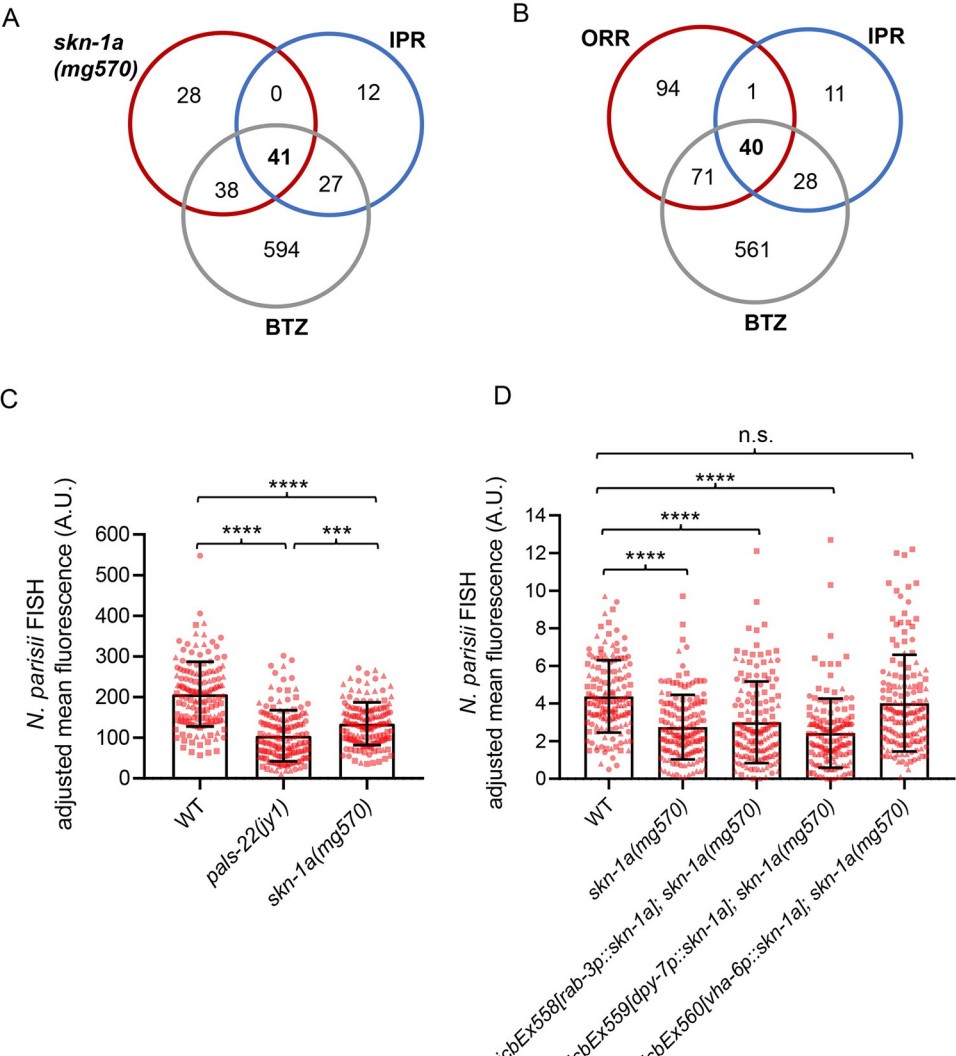

**Fig 4. Proteasome impairment in the intestine leads to activation of the IPR. (A)** Venn diagram showing significant overlap of IPR with up-regulated genes in *skn-1a* mutants (RF 84.4, *p* < 1.790e-72) and IPR with BTZ-treated animals (RF 21.4, *p* < 8.653e-84). **(B)** Venn diagram showing that overlap between ORR and IPR involves genes induced upon BTZ treatment. **(C)** *skn-1a(mg570)* mutants display increased resistance to *N. parisii* at 30 hpi relative to WT animals when infected at young adult. The *pals-22(jy1)* mutant was used as a positive control for its known increased resistance to *N. parisii*. **(D)** Intestinal rescue of *skn-1a*, but not epidermal or neuronal rescue, restores the resistance of *skn-1a(mg570)* mutants to *N. parisii* infection to WT levels. Kruskal–Wallis test with Dunn's multiple comparisons test was used for statistical analysis (**** *p* < 0.0001, *** *p* < 0.001. *n* = 150 animals per genotype). The numerical data for all 3 replicates of panels C and D is available in Supporting information S1 Data. BTZ, bortezomib; IPR, intracellular pathogen response; ORR, oomycete recognition response.

*Nematocida parisii* and the Orsay virus [4,5]. To determine if mutants in the proteasome surveillance pathway also show constitutive activation of the IPR, we compared RNAseq datasets generated for *skn-1a* mutants and BTZ-treated animals with the IPR gene list. We found significant overlap between IPR and up-regulated genes in *skn-1a* mutants (Fig 4A and 4B and S1 Table). To investigate whether loss of *skn-1a* also leads to increased resistance against intestinal pathogens, we assayed the *skn-1a(mg570)* mutant for resistance against the intestinal pathogen *N. parisii*. Here, we found that *skn-1a(mg570)* mutants had increased resistance to *N. parisii* (Fig 4C), which was rescued in this case specifically by intestinal (*vha-6p*) expression of *skn-1a*

(Fig 4D). These results demonstrate that impairment of the SKN-1A proteasome surveillance pathway also induces resistance to intracellular pathogens of the intestine, and this impairment can be rescued specifically by SKN-1A function in the intestine.

We have previously reported that approximately half of the IPR genes are also present in the ORR list [3], which was rather surprising given the distinct infection strategies and tissue tropism between oomycetes and intestinal-infecting microsporidia [42]. We hypothesized that the IPR and ORR programs might share significant overlap because they both involve regulation by the proteasome. Remarkably, we observed that almost all common ORR and IPR genes (40 out of 41 genes) were also regulated by BTZ treatment (Fig 4B and S1 Table). We reasoned that these shared immune response genes may be induced in different tissues following an ORR or IPR trigger. To test this possibility, we made use of single molecule fluorescence in situ hybridization (smFISH) to determine in which tissue selected genes present in the overlap between ORR and IPR are induced, namely, *pals-5*, *B0507.8*, *skr-3*, and *cul-6*. Animals carrying GFP-labeled epidermal nuclei (*dpy-7p*::*GFP-H2B*) were treated with oomycete extract to activate the ORR and with prolonged heat stress (24 h at 30˚C) [5] to induce the IPR. These 2 triggers were chosen in lieu of active infections because they induce responses in a more consistent manner across a population. Treatment with BTZ was performed to simultaneously activate both ORR and IPR. Consistent with our initial hypothesis, we found that all genes were induced specifically in the epidermis upon activation of ORR and in the intestine upon activation of IPR, while BTZ treatment led to induction in both tissues (Figs 5 and S6).

Previous studies have linked *skn-1* to immunity against bacterial infection with *skn-1* loss-of-function mutants reported as being hypersusceptible to bacterial infection [23–25]. However, previous studies were performed with RNAi or mutants that affect multiple isoforms of *skn-1*, and so it has not been clear whether *skn-1a* specifically is involved in bacterial immunity. We investigated this question by examining *skn-1a(mg570)* animals for increased susceptibility to *P. aeruginosa* (PA14) infection [43]. In this assay, *pals-22(icb89)* mutants and *skn-1* RNAi-treated animals were used as positive controls, as both have been shown to result in enhanced susceptibility towards PA14 infection [16,24]. The susceptibility of *skn-1a(mg570)* animals to PA14 was found to be comparable to wild-type animals, while both *skn-1* RNAi and *pals-22* mutants showed increased susceptibility as expected (S7 Fig). The fact that *skn-1* RNAi targets both *a* and *c* isoforms, but *skn-1a(mg570)* animals are not hypersusceptible to PA14 suggests that the requirement of SKN-1 to combat PA14 infection is either associated with the function of SKN-1C or both isoforms have redundant roles in this case. Taken together, these results suggest that different *skn-1* isoform perturbations could lead to different host immunity outcomes in a pathogen-specific way.

## Discussion

We demonstrate in this study that mutations in the SKN-1A proteasome surveillance pathway in *C. elegans* activate the ORR and IPR immune responses employed against distinct natural pathogens that infect through the epidermis or colonize the intestine. Our previous work indicated that blockade of the proteasome leads to induction of IPR genes, which are distinct from SKN-1-regulated genes [4,16], but the connection between the SKN-1-regulated pathway and the ORR/IPR was not clear. Here, we show that both loss of SKN-1A, an ER-localized transcription factor, and impairment of proteasome function induce the genes in common to both ORR and IPR. Furthermore, increasing proteasome function through constitutive proteasomal subunit gene expression or hyperactivity of the proteasome partially inhibits oomycete extract-mediated induction of ORR signature genes. Taken together, we hypothesize that some proteasomally regulated factor(s) normally keep the ORR and IPR immune responses in check in the

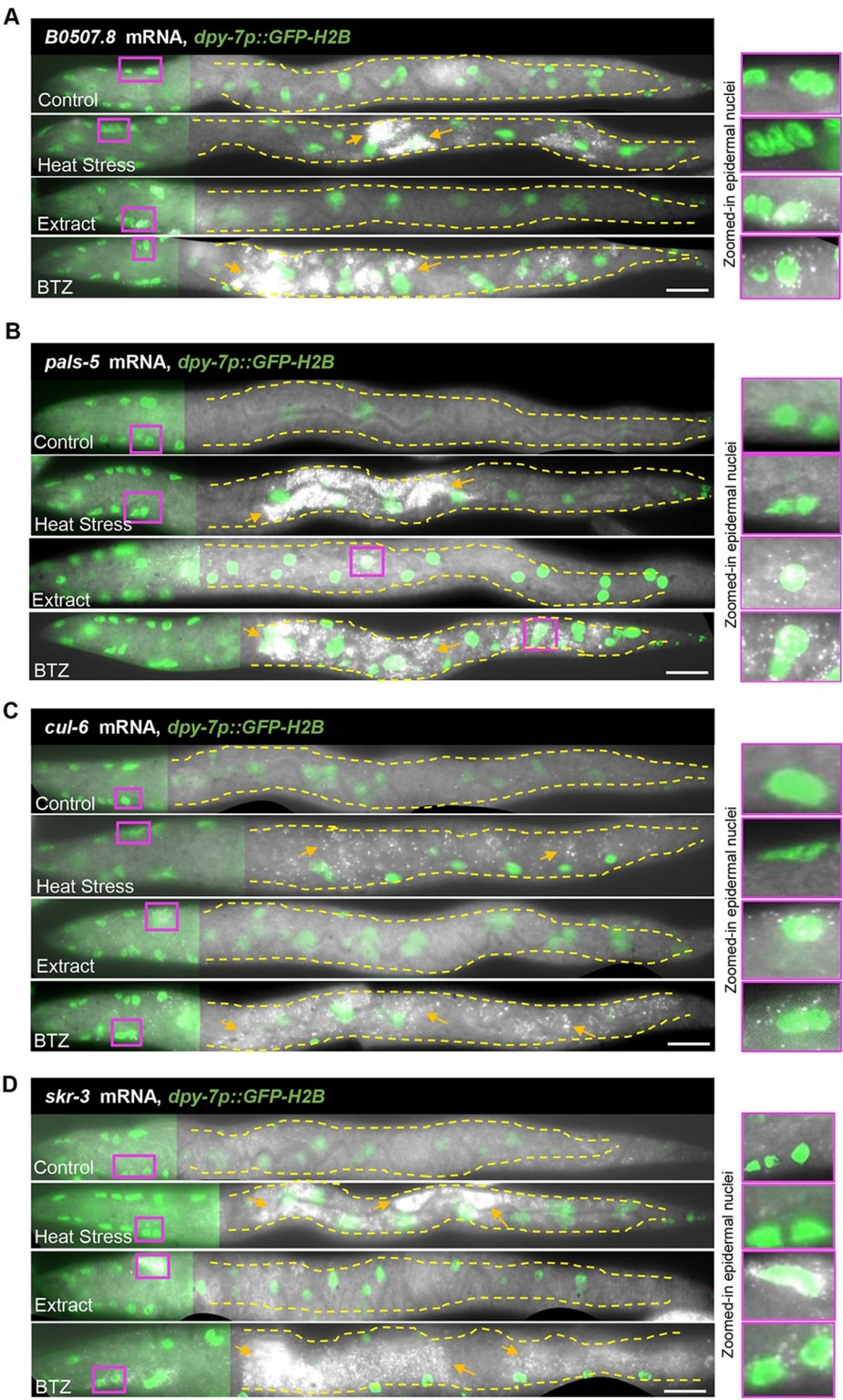

**Fig 5. Common ORR and IPR genes can be induced in a tissue-specific manner.** Sections of straightened L2 stage animals showing mRNA distribution of some common ORR and IPR genes, namely, *B0507.8* (**A**), *pals-5* (**B**), *cul-6* (**C**), and *skr-3* (**D**) by smFISH upon 4 h extract treatment (ORR), prolonged heat stress at 30°C for 24 h (IPR) and 2 h of BTZ treatment (proteasome dysfunction). Epidermal nuclei are labeled in green with the *dpy-7p::GFP-H2B* marker. Co-localization of mRNA with green nuclei indicates epidermal expression as shown in zoomed-in panels (shown in

magenta). Dashed yellow lines outline the intestine and orange arrows point to smFISH signal. Images are presented so that epidermal nuclei in the head region are in focus. The intensity of the GFP signal from out of focus epidermal nuclei in the posterior part of the body is reduced to highlight the signal in the intestine. Scale bar is 10 μm. See S6 Fig for quantification. BTZ, bortezomib; IPR, intracellular pathogen response; ORR, oomycete recognition response; smFISH, single molecule fluorescence in situ hybridization.

absence of infection, and upon exposure to specific pathogens such factors can be stabilized in the associated tissue triggering the rapid induction of the respective immune programs (Fig 6). A similar phenomenon has been shown to regulate activation of NF-κB in *Drosophila*, where IMD is proteasomally degraded owing to permanent presence of $Ub^{K48}$ linkages, which are lost upon bacterial infection thereby stabilizing the protein and activating the pathway [26,27]. Likewise, Uba1-mediated proteasomal degradation of IRF3 in Zebrafish keeps activation of IFN signaling and activation of anti-viral immune response in check [44]. While our findings are consistent with a potential direct role for the proteasome in stabilizing factors that may be necessary for ORR/IPR induction, we cannot rule out that the bounce-back response pathway may also act in parallel. Given that the activation of the *chil-27p*::*GFP* marker is observed only in late larval stages in *skn-1a* mutants or with high doses of BTZ in wild-type animals, we speculate that the extent of proteasome dysfunction may determine the level of activation of the immune response programs. Furthermore, SKN-1A has been recently shown to be involved in lipid homeostasis so it is likely to also influence host immunity in more complex ways and beyond its role in the regulation of proteasome function [45].

Unwanted activation of immune responses is often accompanied by trade-offs emphasizing the importance of keeping them in check [46]. For example, in *pals-22* loss-of-function

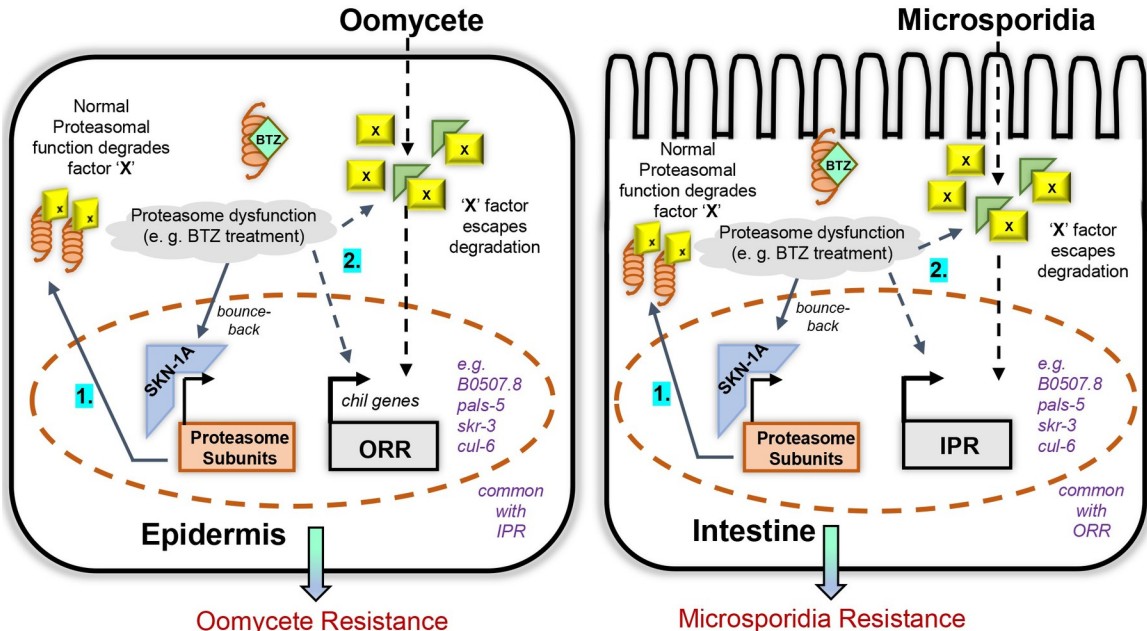

**Fig 6. Model based on the findings of this study.** The model highlights the interplay of SKN-1A-mediated proteasomal gene expression and activation of ORR and IPR in the epidermis and the intestine, respectively. **1.** Under normal conditions, proteasomal degradation of unknown positive regulators (denoted as X factor) keeps activation of immune responses in check. **2.** When proteasome dysfunction happens either by BTZ treatment or loss-of-function of *skn-1a*, X-factor escapes degradation and activates ORR and IPR in the epidermis and the intestine. Simultaneously, BTZ-mediated inhibition increases proteasomal gene expression as a bounce-back response. Such X factors could be directly involved in the tissue-specific signaling pathway activated upon oomycete or microsporidia exposure, respectively, in the epidermis and the intestine. Alternatively, they could regulate ORR and IPR gene induction in parallel to the pathogen-induced signaling pathway. BTZ, bortezomib; IPR, intracellular pathogen response; ORR, oomycete recognition response.

mutants, constitutive activation of ORR/IPR provides them protection from oomycetes and microsporidia [3]. However, *pals-22* mutants also exhibit slow development, reduced lifespan, and increased pathogen load upon exposure to *P. aeruginosa* [16]. Likewise, *pals-17* loss-of-function mutants exhibit constitutive activation of IPR and immunity to intracellular pathogens, but impaired development and reproduction [47]. While *skn-1a(mg570)* mutants develop normally like wild-type *C. elegans*, they too exhibit increased age-associated protein aggregation and consequently have a reduced lifespan [34], so similar trade-offs may also come into play in this case and involve proteotoxicity-driven defects that may exacerbate upon stressful conditions. Likewise, human patients with proteasome-associated autoinflammatory syndromes (PRAAS) show hyperactivation of IFN signaling, but often suffer with severe neurodevelopmental anomalies and skeletal defects [28,29,48].

Our study also highlights the importance of isoform-specific functions for *skn-1*. While *skn-1b* is neuronally expressed and regulates satiety and metabolic homeostasis [49], *skn-1a* and *skn-1c* are likely to have distinct immunomodulatory functions acting in a pathogen-specific way. Previous studies that reported a requirement of *skn-1* for bacterial resistance used either *skn-1* RNAi or *skn-1* mutant alleles that affected both *a* and *c* isoforms [24,25]. However, we now show that *skn-1a* mutants show enhanced resistance to infection by eukaryotic natural pathogens and do not show enhanced susceptibility to PA14, which suggests that bacterial immunity is likely to be driven by SKN-1C or that SKN-1A/C isoforms might have redundant roles in this case. This result also aligns with the fact that just like NRF2 in humans, SKN-1C regulates response to oxidative stress in *C. elegans* [22], and bacterial pathogens have been shown to trigger ROS production in the host [23].

While proteasome-mediated suppression of inflammatory/immune responses might be an evolutionarily conserved phenomenon, the exact mechanisms of how immune responses are kept in check are likely to differ. For example, activation of type-I interferon signaling in mammals as a consequence of proteasome dysfunction has been attributed to ER stress and activation of the unfolded protein response (UPR) [28,50,51]. This is unlikely to be the case in the context of oomycete recognition as we have previously shown that introducing ER stress does not induce *chil-27p::GFP* [2]. Similarly, markers of proteotoxic stress are not induced as a part of the ORR/IPR programs, which makes them distinct from other stress-induced responses. Whether proteasomal regulation of the same factors keeps immune responses in check in the *C. elegans* epidermis and intestine is currently unknown. In the epidermis, the receptor tyrosine kinase OLD-1 is fully required for mounting the ORR and is likely regulated by the proteasome [39]; however, *old-1* knockouts are still able to mount the ORR upon *skn-1a* perturbation ruling out OLD-1 as the main driving factor. In the intestine, ZIP-1 is a plausible candidate because it is degraded by the proteasome under basal conditions and has been shown to be required for induction of a subset of IPR genes [52]. Future work will elucidate the exact proteasomally regulated triggers of ORR/IPR and address whether these are shared or not between different tissues in *C. elegans*.

## Materials and methods

### *C. elegans* strains and pathogen maintenance

*C. elegans* strains were cultured on NGM plates seeded with *E. coli* OP50 at 20°C under standard conditions as previously described [53]. The list of all the strains used in this study is provided in S2 Table. We refer to *skn-1(mg570)* mutants throughout the manuscript as *skn-1a (mg570)* to stress that only isoform *a* is affected. Both *M. humicola* and *N. parisii* were maintained as described previously [2,42].

### EMS mutagenesis

L4 stage WT animals carrying the *chil-27p*::GFP reporter (*icbIs4*) were mutagenized with 24 mM ethyl methanesulfonate (EMS) (Sigma) in 4 ml M9, for 4 h with intermittent mixing by inversion. Worm pellets were then washed 10 times with 15 ml M9 to completely get rid of EMS from the suspension and worms were plated onto 90 mm NGM plate seeded with *E. coli* OP50. After 24 h, 300 adults were randomly picked and divided into thirty 90 mm plates carrying 10 animals in each. After 72 h at 20°C, all F1 gravid adults in each plate were individually bleached and respective pool of F2 embryos were collected onto new plates. Around 60,000 haploid genomes were screened to identify animals showing constitutive activation of *chil-27p*::GFP. The causative mutations were mapped to *ddi-1* and *png-1* by crossing to the highly polymorphic CB4856 strain, and 15–25 F2 recombinants were pooled in equal proportion to obtain genomic DNA for whole-genome sequencing of each independent mutant, performed by BGI (Hong Kong). WGS data was analyzed using the CloudMap Hawaiian variant mapping pipeline to identify causative mutations [54].

### Oomycete *M. humicola* infection assays

Five *M. humicola* infected (dead) animals were added to the lawn of *E. coli* OP50 on 3 NGM plates and 20 live L4 animals for each genotype were transferred individually to each plate (*n* = 60 per condition per repeat). Dead animals with visible sporangia were scored every 48 h and live animals were transferred to a new NGM plate with 6 *M. humicola* infected (dead) animals. Dead animals without evidence of infection or animals missing from the plate were censored. Infection assays were performed in triplicates at 20°C on 30 mm NGM plates seeded with 100 µl *E. coli* OP50. For infection assays upon *skn-1* RNAi, embryos were collected by bleaching gravid adults and were grown on *E. coli* HT115 (control RNAi) or *skn-1* RNAi bacteria until the adult stage, which were then used to perform infection assay as described above. Similarly, embryos from WT animals were grown on *E. coli* OP50 until early L4 stage and then were transferred to OP50 plates with 20 µm of BTZ (Sigma) or DMSO (Thermo Scientific) (control, <0.1%) for 24 h before using them for infection assay.

### *P. aeruginosa* PA14 infection assays

A total of 100 µl of overnight grown PA14 culture in LB was spread on a 55 mm NGM plate containing 10 µg/ml FuDR. The plates were incubated at 37°C for 24 h, and 90 Day 1 adults grown on NGM-OP50 or NGM-HT115/*skn-1* RNAi in the presence of FuDR were transferred to 3 plates each containing 30 animals (in triplicates) and incubated at 25°C. The number of animals alive was counted every 24 h and dead animals defined as unresponsive to touch were removed from the plate. The experiment was continued until all animals on the plate were dead.

### Microsporidia *N. parisii* infection assays

The 600 synchronized L1 stage worms for each genotype were plated to NGM + *E. coli* OP50-1 and grown to the young adult life stage at 23°C (48 h for all stains except for *pals-22(jy1)* mutants, which are developmentally delayed [5] and therefore were grown for 52 h). The 600 young adults were washed off the plate with M9 + 0.1% Tween-20 (M9-T), pelleted, and the supernatant was removed. The worms were then mixed with 2 million *N. parisii* spores (strain ERTm1), 50 µl of 10× concentrated *E. coli* OP50-1, and M9 to a final volume of 300 µl. The infection mix was top-plated over the entire surface of a 6-cm NGM plate, dried at room temperature, and transferred to 25°C for 3 h. Following the 3 h pulse infection, the worms were

collected from the infection plate with M9-T into a 1.5 ml microcentrifuge tube, pelleted, and then washed 3 additional times with 1 ml of M9-T. The infected worms were then transferred to an NGM + *E. coli* OP50-1 plate without *N. parisii* spores and returned to 25°C for an additional 27 h. Infected animals were then fixed in 100% acetone and incubated at 46°C overnight with FISH probes conjugated to the red Cal Fluor 610 fluorophore that hybridize to *N. parisii* ribosomal RNA (Biosearch Technologies). Samples were analyzed for *N. parisii* meronts [55] using an ImageXpress Nano plate reader using the 4× objective (Molecular Devices, LLC). The worm area was traced using FIJI software and the average red fluorescence intensity of each worm was quantified with the background fluorescence of the well subtracted, and 50 animals per genotype were quantified for each experimental replicate, and 3 independent infection experiments were performed.

For the tissue-specific *skn-1a* rescue strains MBA1660, MBA1661, and MBA1662, approximately 100 *bus-1p::GFP*-expressing worms were manually picked onto *N. parisii* infection assay plates. For consistency, approximately 100 adults for WT and *skn-1a(mg570)* controls were also picked in the same manner and pulse infection was performed as described above. To remove any potential bias, the order in which strains were picked was randomized from assay to assay, and the 30 hpi endpoints were staggered to account for picking time. Fixation and FISH hybridization were performed as described above. For quantification, fixed animals for each strain were mounted to a 5% agarose pad on a microscope slide and imaged on a Zeiss AxioImager M1 compound microscope using a 2.5× objective. The worm area was traced using FIJI software and the average red fluorescence intensity of each worm was quantified with the background fluorescence of the slide subtracted. In some samples, we observed dim red autofluorescence from embryos inside the adult gonad. Thus, this region of the worm was omitted in all samples when tracing the worm area. Analysis with the ImageXpress plate reader and Zeiss AxioImager produced different arbitrary values for *N. parisii* FISH pixel intensity. However, the fold difference in infection between WT animals and *skn-1a(mg570)* mutants was similar (WT animals are approximately 1.5-fold more infected than *skn-1a(mg570)*) by both imaging and quantification methods, and 50 animals per genotype were quantified for each experimental replicate, and 3 independent infection experiments were performed.

## RNAseq

For transcriptome analysis, synchronized L4 stage animals were collected in triplicates for RNA extraction using TRIzol (Invitrogen) and isopropanol/ethanol precipitation. RNA sequencing was completed by BGI (Hong Kong). Kallisto [56] was used for alignments with the WS283 transcriptome from Wormbase. Count analysis was performed using Sleuth [57] along with a Wald test to calculate $\log_2$fold changes. All RNAseq data files are publicly available from the NCBI GEO database under the accession number GSE241087.

## RT-qPCR

Synchronized L4 stage animals with 4-h extract treatment or without were collected in TRIzol (Invitrogen) followed by RNA extraction using isopropanol/ethanol precipitation. RNA was quantified using NanoDrop (Thermo Scientific) and its quality was analyzed by gel electrophoresis. cDNA was synthesized using 2 µg RNA with Superscript IV (Invitrogen) and Oligo(dT) primers as per manufacturer's instructions. Real-time PCR was performed using qPCR primer pairs listed in S2 Table and LightCycler480 SYBR Green I Master Mix (Roche) in a LightCycler480 instrument, and Ct values were derived using the LightCycler480 software and second derivative maximum method. All experiments were performed in biological triplicates and changes in expression were calculated via the 2-ΔΔCt method.

## Microscopy

Animals to be imaged were picked into a drop (5 to 7 µl) of M9 containing sodium azide (50 µm) on an agarose (1%) pad made on a glass slide. Coverslip was added once the animals were paralyzed, and the pad was almost dry. *chil-27p::GFP/col-12p::mCherry* expression was observed on Zeiss Axio Zoom V16 microscope and images were taken using the associated ZEN Microscopy software. For *rpt-3p::GFP* and *sur-5::UbV-GFP*, animals were treated with oomycete extract or 20 µm BTZ with DMSO control for 24 h at early L2 stage and expression was observed on Zeiss Compound microscope (AxioScope A1) at 40× magnification and images were taken using OCULAR. For *chil-27p::GFP* expression upon BTZ treatment, early L4 stage animals were treated with different doses of BTZ for 24 h and adults were imaged as described above.

## Molecular cloning and transgenesis

To generate constructs for tissue-specific rescue of *skn-1a(mg570)* mutant, *skn-1a* fragment was amplified from N2 cDNA using primers skn-1a_fullF and skn-1a_fullR. Promoter fragments, namely, *rab-3p* and *vha-6p* were amplified from N2 genomic DNA using primer pairs BJ97-pRab-3Fwd, rab-3rev-skn1, bj97 vha-6 Fwd, and vha-6 Rev skn-1a, respectively. The terminator sequence from 3′UTR of *unc-54* was PCR amplified from N2 genomic DNA using primers unc-54-F-skn1a and BJ36_unc-54terR. Plasmids for neuronal and intestinal rescue were assembled with specific promoter, *skn-1a* and *unc-54* 3′UTR fragments ligated into SpeI (FastDigest)-digested pBJ97 by Gibson cloning. To make the epidermal rescue plasmid, *skn-1a* was PCR amplified from N2 cDNA using primers dpy-7_skn-1F and dpy-7_skn-1R and was assembled into PmeI(FastDigest)-digested pIR6 by Gibson cloning. All constructs were individually injected into *skn-1a(mg570)* at 5 ng/µl with 25 ng/µl of pRJM163 (*bus-1p::GFP*) as the co-injection marker and 80 ng/µl of pBJ36 as carrier DNA. The transgenic strains thus created were used for microsporidia pathogen load assays. Similarly, all constructs were individually also injected in the strain MBA1055, which is *skn-1a(mg570)* mutant with *chil-27p::GFP* reporter in the background. For these injections, 5 ng/µl of rescue plasmid with 5 ng/µl of *myo-2p::GFP* as co-injection marker and 100 ng/µl of pBJ36 as carrier DNA was used. The transgenic strains thus created were analyzed for expression of *chil-27p::GFP* in *skn-1a(mg570)* mutants.

## RNAi

RNAi by feeding was used as the means to induce gene knockdown in *C. elegans* as previously described [58]. The RNAi clone for *skn-1* was obtained from the ORFeome Library (Horizon Discovery) [59] and for *wdr-23* and *rpt-5* were obtained from the Ahringer Library (Source BioScience) [60]. Both the clones were confirmed by sequencing prior to use. Embryos collected by bleaching gravid adults were added onto NGM plates seeded with *E. coli* HT115 and contained 1 mM IPTG (Sigma) to induce expression of dsRNA. After 72 h of incubation at 20°C, animals at L4/adult stage were scored for *chil-27p::GFP* expression ($n > 50$). The strains used for tissue-specific RNAi were generated by introducing the *chil-27p::GFP* reporter in previously described strains listed in S2 Table with their associated references.

## smFISH

For inducing the ORR, synchronized L2 stage animals were treated with oomycete extract for 4 h. For inducing IPR, animals were subjected to prolonged heat stress by incubating synchronized L1s at 30°C for 24 h. For causing proteasome dysfunction, synchronized L2 stage

animals were treated with 20 μm BTZ treatment for 15 min or 2 h. Following requisite treatment along with relevant controls, animals were fixed with 4% formaldehyde (Sigma-Aldrich) in 1× PBS (Ambion) for 45 min and were permeabilised with 70% ethanol for 24 h. Hybridization was performed at 30°C for 16 h. List of oligos included in all the probes can be found in S2 Table. Imaging was performed in an inverted and fully motorized epifluorescence microscope (Nikon Ti-eclipse) with an iKon M DU-934 CCD camera (Andor) controlled via the NIS-Elements software (Nikon) using the 100× objective. Detailed protocol can be found in [61].

## Supporting information

**S1 Fig. Proteasome surveillance mutants and *skn-1* RNAi show constitutive *chil-27p*::GFP expression. (A)** Gene structure of *ddi-1* and *png-1* showing the positions of mutant alleles used in this study. **(B)** Quantification of induction of *chil-27p*::GFP in the mutants obtained from the EMS screen, *ddi-1(icb156)*, *png-1(icb112)*, *png-1(icb121)* along with active-site mutant of *ddi-1(mg572)*, in-frame gene deletion of *ddi-1(mg571)*, *png-1(ok1654)*, *skn-1a(mg570)* and upon *skn-1* RNAi ($n > 50$, ****$p$-value $< 0.0001$, *** $p$-value $< 0.001$ based on chi-square test). The numerical data for all 3 replicates is available in Supporting information S1 Data. **(C)** Venn comparisons showing significant overlap between up-regulated genes in the transcriptome of *ddi-1(icb156)* and *skn-1a(mg570)* mutant with bortezomib (BTZ)-treated animals (RF 6.0, $p < 4.347e-43$ and RF 18.6, $p < 4.453e-88$, respectively).
(TIF)

**S2 Fig. Effect of loss or gain of SKN-1C function on *skn-1a(mg570)* mutants. (A)** Gene structure of *skn-1a* and *skn-1c* isoforms with protein domain organization. **(B)** Both loss (*skn-1* RNAi) or gain (*wdr-23* RNAi) of SKN-1C function does not affect *chil-27p*::GFP expression in *skn-1a(mg570)* (control RNAi) animals. *skn-1a(mg570)* animals are sensitive to RNAi as shown by the *gfp* RNAi treatment leading to complete loss of *chil-27p*::GFP expression (left panel). *wdr-23* RNAi activates SKN-1C and leads to expression of *gst-4p*::tdTomato as shown in the right panel. ($n > 50$ per condition, performed in triplicates, representative image shown). Scale bar is 100 μm. **(C)** Survival curve of WT and *skn-1a(mg570)* animals treated and untreated with *skn-1* RNAi in the presence of the oomycete *M. humicola* ($n = 60$ per condition, performed in triplicates, $p < 0.001$ based on log-rank test, a representative graph for one of the 3 replicates is shown). The numerical data for all 3 replicates is available in Supporting information S1 Data.
(TIF)

**S3 Fig. Bortezomib (BTZ) treatment induces *chil-27* expression. (A)** *chil-27p*::GFP expression in WT animals treated with different doses of BTZ at early L4 stage for 24 h ($n > 50$ for each condition, performed in triplicates, representative image shown). **(B)** Z-stack of L2 stage WT animals showing expression of *chil-27* mRNA upon 15 min and 1 h post extract and BTZ treatment. Scale bar is 100 μm in (A) and 10 μm in (B). **(C)** Survival curve of WT animals treated with DMSO or 20 μm BTZ in the presence of the oomycete *M. humicola* ($n = 60$ per condition, performed in triplicates, $p < 0.001$ based on log-rank test, a representative graph for one of the 3 replicates is shown). The numerical data for all 3 replicates of panels S3A and C is available in Supporting information S1 Data.
(TIF)

**S4 Fig. Epistasis analysis of *skn-1* and other regulators of the ORR.** *skn-1* RNAi induces *chil-27p*::GFP expression in *pals-25(jy81)* and *old-1(ok1273)* mutant animals. Scale bar for all panels is 100 μm, and $n > 50$ per condition, performed in triplicates, representative image

shown.
(TIF)

**S5 Fig. RT-qPCR for selected ORR genes in extract-treated, untreated, and *skn-1a(mg570)* mutants.** In the absence of extract treatment in animals with constitutive expression of the activated form of SKN-1A[*skn-1a(cut, 4ND)* in *skn-1a(mg570)*] or constitutive activation of the proteasome [*pas-3(α3ΔN)*] no expression of ORR genes was observed similar to wild-type animals. Note that ORR gene expression in *skn-1a(mg570)* mutants is lower compared to extract-treated wild-type animals (*$p < 0.05$, **$p < 0.01$, ****$p < 0.0001$ based on unpaired *t* test). The numerical data for all 3 replicates is available in Supporting information S1 Data.
(TIF)

**S6 Fig. Quantification of expression of common ORR and IPR genes shown in Fig 5.** Whole animal mRNA quantification by smFISH for 4 genes, namely, *B0507.8* (**A**), *pals-5* (**B**), *cul-6* (**C**), and *skr-3* (**D**) in L2-stage animals subjected to prolonged heat stress (30˚C for 24 h), extract treatment (4 h), and 20 μm BTZ treatment (2 h) (*n* = 12 in each condition and one-way ANOVA and Tukey's multiple comparison test was used to assess statistical significance, *$p < 0.05$, **$p < 0.01$, ***$p < 0.001$, ****$p < 0.0001$). The numerical data for graphs A–D is available in Supporting information S1 Data.
(TIF)

**S7 Fig. *skn-1a(mg570)* mutants do not show hyper-susceptibility to *P. aeruginosa* PA14.** Survival analysis of adult *C. elegans* on PA14 where bacteria was spread on the plate (*n* = 90 per condition, performed in triplicates, $p < 0.001$ based on log-rank test, a representative graph for one of the 3 replicates is shown). The numerical data for all 3 replicates is available in Supporting information S1 Data.
(TIF)

**S1 Table. Differentially expressed genes in *ddi-1(icb156)* mutants as compared to WT animals, along with datasets used to make Venn diagrams, and the list of overlapping genes in different conditions.**
(XLSX)

**S2 Table. List of strains, oligos, and smFISH probes used in the study.**
(XLSX)

**S1 Data. Numerical data for all graphs presented in the study.**
(XLSX)

## Acknowledgments

We thank Domenica Ippolito and Kenneth Liu for comments on the manuscript. Some strains were provided by the CGC, which is funded by NIH Office of Research Infrastructure Programs (P40 OD010440). We thank Gary Ruvkun, Thorsten Hoppe, Keith Choe, and David Smith for strains.

## Author Contributions

**Conceptualization:** Manish Grover, Emily R. Troemel, Michalis Barkoulas.

**Data curation:** Manish Grover, Spencer S. Gang, Emily R. Troemel, Michalis Barkoulas.

**Formal analysis:** Manish Grover, Spencer S. Gang, Emily R. Troemel, Michalis Barkoulas.

**Funding acquisition:** Emily R. Troemel, Michalis Barkoulas.

**Investigation:** Manish Grover, Spencer S. Gang.

**Methodology:** Manish Grover, Spencer S. Gang.

**Resources:** Emily R. Troemel, Michalis Barkoulas.

**Supervision:** Emily R. Troemel, Michalis Barkoulas.

**Validation:** Emily R. Troemel, Michalis Barkoulas.

**Visualization:** Emily R. Troemel, Michalis Barkoulas.

**Writing – original draft:** Manish Grover, Michalis Barkoulas.

**Writing – review & editing:** Manish Grover, Spencer S. Gang, Emily R. Troemel, Michalis Barkoulas.

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
