## [Editor Report · Decision Letter 0]

1 Sep 2023

Dear Dr. Barkoulas, 

Thank you for submitting your manuscript entitled "Impairment of the SKN-1A/NRF1 proteasome surveillance pathway triggers tissue-specific protective immune responses against distinct natural pathogens in C. elegans." for consideration as a Research Article by PLOS Biology.

Your manuscript has now been evaluated by the PLOS Biology editorial staff, as well as by an academic editor with relevant expertise, and I am writing to let you know that we would like to send your submission out for external peer review.

Once your full submission is complete, your paper will undergo a series of checks in preparation for peer review. After your manuscript has passed the checks it will be sent out for review. To provide the metadata for your submission, please Login to Editorial Manager (https://www.editorialmanager.com/pbiology) within two working days, i.e. by Sep 03 2023 11:59PM.

Kind regards,

Paula

---

Senior Editor

PLOS Biology

---

## [Decision Letter · Decision Letter 1]

12 Oct 2023

Dear Dr. Barkoulas,

Thank you for your patience while your manuscript "Impairment of the SKN-1A/NRF1 proteasome surveillance pathway triggers tissue-specific protective immune responses against distinct natural pathogens in C. elegans." went through peer-review at PLOS Biology. Your manuscript has now been evaluated by the PLOS Biology editors, an Academic Editor with relevant expertise, and by several independent reviewers.

In light of the reviews, which you will find at the end of this email, we are pleased to offer you the opportunity to address the comments from the reviewers in a revision that we anticipate should not take you very long. We will then assess your revised manuscript and your response to the reviewers' comments with our Academic Editor aiming to avoid further rounds of peer-review, although might need to consult with the reviewers, depending on the nature of the revisions.

In particular, the academic editor and us think that the requests from reviewer #2 are not necessary for publication. We consider important that you address the issues from reviewer #3, and please also address the issues from reviewer #1. 

**IMPORTANT - SUBMITTING YOUR REVISION**

*Resubmission Checklist*

*Published Peer Review*

*PLOS Data Policy*

*Blot and Gel Data Policy*

Sincerely,

Paula

---

Senior Editor

PLOS Biology

REVIEWS:

Reviewer #1: C elegans immunity.

Reviewer #2: C elegans gene regulation.

Reviewer #3: Proteasome misregulation.

Reviewer #1: This paper is full of superb surprises. First the screen for the genetic pathway that activates a chitinase response to an oomycete infection surprisingly finds two of the 3 components of the skn-1a proteasomal response pathway. It demonstrates how a great GFP screen can rather instantly tell the authors what pathway they are studying. The involvement of png-1 and ddi-1 and skn-1a was not AT ALL predictable from first principles of innate immunity. So the authors find themselves in a superb intersection between semi mature fields. And it is highly interesting that proteasome homeostasis is a central player in response to oomycete infection.

The rescue of the skn-1a resistance in hypodermis in figure 3 is convincing.

The smFish in Figure 5 was really difficult for me to discern. It is not a color blind issue. Perhaps better labeling would "sell" the commonality of the response to the various pathogens and BTZ.

This is the main conclusion and it is well proven by the experiments in Figures 1 to 5: "we hypothesize that some proteasomally regulated

320 factor(s) normally keep the ORR and IPR immune responses in check in the absence

321 of infection, and upon exposure to specific pathogens such factors can be stabilized

322 in the associated tissue triggering the rapid induction of the respective immune

323 programmes (Figure 6)."

SKN-1A is one of the rare transcription factors that goes to the ER, an organelle at the center of secreted protein responses to pathogens. So the finding that skn-1a is at the center of ORR is really cool. Perhaps another sentence in the discussion to get the readers up to speed. 

Small points

1. Figure 1C is really supplemental....showing locations of alleles is only worthy of a main display figure if there is something interesting about locations or types of mutations. 

2. Figure 1E Y axis should be labelled with "% survival w oomycete infection" (always aim to make Figures interpretable without having to read Figure legends for key hints).

3. Figure2C is not visible to the Red Green Color blind without activating color blind accessibility triggers.

Reviewer #2: This interesting manuscript describes a homeostatic relationship between two responses to eukaryotic pathogens and the proteasome, with a key role being played by SKN-1A/Nrf1, the master transcriptional regulator of proteasome subunit gene transcription. The authors uncovered this relationship through a genetic screen that led them to regulators of SKN-1A. The results are of interest not only to the pathogenesis and immunity fields but also generally, because they uncover a pathway of crosstalk between immune defenses and the ubiquitin proteasome system. An additional benefit is that the paper defines a new example of evidence that discriminates between the functions of SKN-1A/Nrf1 and SKN-1C/Nrf2, which are expressed from alternatively spliced isoforms of the same gene. Studies in this area frequently conflate or confuse the functions of these two proteins, which are each important in aging, stress resistance, metabolism, and now immunity. The manuscript is well-written and the data are of high quality, with an important exception noted below.

Major comment:

The time has past (or should be past) when it is sufficient to present pictures of grouped worms or individual images to assess gene expression. The former is highly subjective and unreliable, and the latter (i.e. Fig. 5) gives us no indication of the numbers analyzed or reproducibility. This is no longer allowed at most top journals. I hesitate to ask the authors to repeat all of these experiments (the ideal situation), but they really should take significant steps in this direction like repeating ANY experiments that are not clear on/off situations with scoring and quantification, statistics. They should also indicate the numbers of animals analyzed and consistency of results for ALL such experiments. In today's world some experiments should be scored blindly, as well.

It is interesting that the authors have implicated proteasomal homeostasis in these immunity pathways, but the paper would have more impact if they determined what regulator(s) was/were targeted by the proteasome. The Troemel lab has implicated the transcription factor ZIP-1 in the IPR but this, oddly, was not mentioned. Could this factor or another candidate be targeted by the proteasome? Sorting this out is not necessarily essential for this paper but would greatly strengthen it.

Other comments:

In addition to the experiments performing wdr-23 RNAi, a complementary approach for ruling out functions for SKN-1C would be to perform skn-1 RNAi in WT and skn-1a mutant strains in parallel. This should be applied to the gene expression and pathogenesis studies, or at least some of them. It is a direct way to address SKN-1C functions that could strengthen the claims made.

A recent study (PMID: 36598980) implicated SKN-1A in fat/lipid metabolism and showed that its absence increases stored fat levels. Given that availability of specific fats has been implicated in other immune responses, this should be considered or at least discussed. Were any SKN-1A-regulated fat metabolism genes picked up in the datasets described here?

As an alternative model, is it known whether the genes involved in these immune responses have conserved recognition sequences that would be bound by SKN-1A and repressed? 

Very minor: The Lehrbach 2016 eLife paper is a landmark in this area and should be cited earlier, in the introduction.

Reviewer #3: The conserved SKN-1A/Nrf1 transcription factor regulates proteasome subunit gene expression to ensure adequate proteasome function. In this study, Grover et al show that mutations that disrupt SKN-1A/Nrf1 in C. elegans lead to misregulation an innate immune signaling pathway called the oomycete recognition response (ORR). Given previous work suggesting that both the ORR, and a related immune response to intracellular pathogens (the IPR), are activated in animals experiencing proteasome dysfunction, this finding prompted the authors to explore the relationship between skn-1, the proteasome, and immunity in C. elegans. 

Overall, this work advances the field by making a novel link between SKN-1/Nrf and innate immune regulation and refines our understanding of the interplay between the proteasome and C. elegans immune regulation that had been hinted by previous studies. The interest to a wider scientific audience is somewhat limited without mechanistic insight into the regulatory mechanism(s) that link proteasome dysfunction to innate immune control, but addressing this issue would be better left to future studies. I recommend that this work is suitable for publication in PLoS Biology with minor changes and limited additional experiments to address the comments listed below.

Comments.

The authors show that skn-1a mutants show increased expression of ORR genes and resistance to oomycete infection and propose that both of these effects are the result of impaired proteasome function in skn-1a mutants. However, SKN-1A/Nrf1 may also regulate other genes aside from the proteasome subunits. The argument that the skn-1a mutants' resistance to oomycete infection results from reduced proteasome function (rather that misregulation of other skn-1a target genes) would be strengthened by testing the effect of BTZ treatment or proteasome subunit RNAi on resistance to oomycete infection.

Related to the above point - knock-down of proteasome subunits by RNAi sensitizes C. elegans to infection by N. parisii (Bakowski et al 2014, PMID: 24945527), suggesting that the proteasome is required for defense against infection by this pathogen. In this study, the authors find that skn-1a mutants (in which proteasome subunit expression is reduced) are resistant to N. parisii infection. It would be interesting for the authors to include a discussion of this discrepancy. One interesting possibility could be that the proteasome has protective roles in combatting infection, in addition to a role in regulating immune responses. In that case, differing degrees of proteasome dysfunction may have different results on immunity. I.e., mild proteasome dysfunction might enhance pathogen resistance through ORR/IPR activation, whereas severe proteasome dysfunction could increase sensitivity by disrupting proteasome-dependent immune defenses. This could be tested by comparing the effects of different BTZ concentrations on C. elegans' sensitivity to infection. 

The evidence that hyperactivation of the proteasome attenuates immune responses (fig 3D) needs to be strengthened and/or interpreted more cautiously:

Firstly, in this experiment the expression of ORR genes in skn-1a(cut, 4nd) and pas-3∆N animals treated with extract is compared to WT animals treated with extract as a control. The authors conclude that: 'hyperactivity of the proteasome significantly inhibits oomycete extract-mediated chil-27 gene induction.' However, the data presented do not distinguish between the possibility that animals with hyperactive proteasomes are defective in induction of chil-27, Vs simply show reduced chil-27 expression levels regardless of exposure to extract. The levels of chil-27 mRNA must be compared between WT and both skn-1a(cut, 4nd) and pas-3∆N in the absence of oomycete extract. If chil-27 mRNA level is identical in the absence of extract, this supports the statement that induction is defective. But if uninduced chil-27 mRNA level is also reduced, this would suggest that proteasome hyperactivation generally reduces chil-27 mRNA levels (but does not necessarily disrupt the pathogen-responsive signaling pathway that induces chil-27 and other ORR genes in response to extract).

Secondly, animals with hyperactive proteasomes (skn-1a(cut, 4nd) and pas-3∆N) express ORR genes at relatively high levels compared to uninduced WT in response to oomycete extract exposure, even if expression levels are slightly lower than in similarly induced WT animals. So it is unclear whether this change in immune gene expression has any functional significance. This should be addressed by measuring the effect of skn-1a(cut, 4nd) and pas-3∆N on survival following oomycete infection.

In Fig 5, prolonged heat-stress was used as a proxy for inducing the IPR, could the authors please explain why this was used instead of pathogen exposure? Since prolonged heat stress would presumably affect all tissues of the animal, why does prolonged heat stress only activate the intestinal IPR but not the hypodermal ORR?

The differing phenotype caused by skn-1(RNAi) Vs the skn-1a mutant is used to infer that the role of skn-1 in PA14 resistance is more likely to be associated with SKN-1C isoform. But these data are equally consistent with the possibility that SKN-1A and SKN-1C isoforms are redundant for this function. The text should be modified to include this possibility.

The idea that different skn-1 isoforms are 'pathogen-specific' should be supported by experiments to address the role of SKN-1C in resistance to eukaryotic pathogens. This could be done by testing whether skn-1(RNAi) - ie kockdown of both skn-1a and skn-1c - alters the resistance of WT and/or the hyper-resistant skn-1a mutants to oomycete or N. parisii infection.

Minor comments.

Typo: On line 103, the NGLY1 gene is referred to as NGLY

NRF1 should also be referred to as NFE2L1 (at least when NRF1 is first mentioned in the introduction at line 99) to avoid confusion with Nuclear Respiratory Factor 1, which can also be referred to by the acronym NRF1.

---

## [Editor Report · Decision Letter 2]

18 Dec 2023

Dear Dr Barkoulas,

Thank you for your patience while we considered your revised manuscript entitled "Impairment of the SKN-1A/NRF1 proteasome surveillance pathway triggers tissue-specific protective immune responses against distinct natural pathogens in C. elegans" for publication as a Research Article at PLOS Biology. This revised version of your manuscript has been evaluated by the PLOS Biology editors and by the Academic Editor.

Based on our Academic Editor's assessment of your revision, we are likely to accept this manuscript for publication, provided you satisfactorily address the data and other policy-related requests stated below.

In addition, we would like you to consider a suggestion to improve the title:

"Proteasome inhibition triggers tissue-specific immune responses against different pathogens in C. elegans"

We expect to receive your revised manuscript within three weeks. 

*Published Peer Review History*

*Press*

Sincerely,

Ines

--

Ines Alvarez-Garcia, PhD

Senior Editor

PLOS Biology

Fig. 1D; Fig. 2C; Fig. 3A, D; Fig. 4C, D; Fig. S1B; Fig. S2C; Fig. S3A, C; Fig. S5; Fig. S6A-D and Fig. S7

---

## [Editor Report · Decision Letter 3]

9 Feb 2024

Dear Dr Barkoulas,

Thank you for the submission of your revised Research Article entitled "Proteasome inhibition triggers tissue-specific immune responses against different pathogens in C. elegans" for publication in PLOS Biology. On behalf of my colleagues and the Academic Editor, Hans-Uwe Simon, I am delighted to let you know that we can in principle accept your manuscript for publication, provided you address any remaining formatting and reporting issues. These will be detailed in an email you should receive within 2-3 business days from our colleagues in the journal operations team; no action is required from you until then. Please note that we will not be able to formally accept your manuscript and schedule it for publication until you have completed any requested changes.

PRESS

Sincerely, 

Ines

--

Ines Alvarez-Garcia, PhD

Senior Editor

PLOS Biology
